# Safety of ChAdOx1 nCoV-19 Vaccine: Independent Evidence from Two EU States

**DOI:** 10.3390/vaccines9060673

**Published:** 2021-06-18

**Authors:** Abanoub Riad, Andrea Pokorná, Mohamed Mekhemar, Jonas Conrad, Jitka Klugarová, Michal Koščík, Miloslav Klugar, Sameh Attia

**Affiliations:** 1Department of Public Health, Faculty of Medicine, Masaryk University, Kamenice 5, 625 00 Brno, Czech Republic; abanoub.riad@med.muni.cz (A.R.); klugarova@med.muni.cz (J.K.); koscik@med.muni.cz (M.K.); 2Czech National Centre for Evidence-Based Healthcare and Knowledge Translation (Cochrane Czech Republic, Czech EBHC: JBI Centre of Excellence, Masaryk University GRADE Centre), Institute of Biostatistics and Analyses, Faculty of Medicine, Masaryk University, Kamenice 5, 625 00 Brno, Czech Republic; apokorna@med.muni.cz; 3Department of Nursing and Midwifery, Faculty of Medicine, Masaryk University, Kamenice 5, 625 00 Brno, Czech Republic; 4Institute of Health Information and Statistics of the Czech Republic, Palackého Náměstí 375/4, 128 01 Praha, Czech Republic; 5Clinic for Conservative Dentistry and Periodontology, School of Dental Medicine, Christian-Albrecht’s University, 24105 Kiel, Germany; mekhemar@konspar.uni-kiel.de (M.M.); conrad@konspar.uni-kiel.de (J.C.); 6Czech Clinical Research Infrastructure Network, Department of Pharmacology, Faculty of Medicine, Masaryk University, Kamenice 5, 625 00 Brno, Czech Republic; 7Department of Oral and Maxillofacial Surgery, Justus-Liebig-University, Klinikstrasse 33, 35392 Giessen, Germany; sameh.attia@dentist.med.uni-giessen.de

**Keywords:** AZD1222, ChAdOx1 nCoV-19, COVID-19, Czech Republic, drug-related side effects and adverse reactions, Germany, health personnel, mass vaccination, Oxford–AstraZeneca vaccine

## Abstract

Recent reports of thrombosis following AstraZeneca COVID-19 vaccine in young females (<55 years-old) led to temporary suspension and urgent investigation by the European Medicines Agency (EMA) that concluded that vaccine benefits still outweigh its side effects (SEs). Therefore, this study aims to provide early independent evidence on the vaccine SEs’ prevalence and their potential risk factors; a cross-sectional survey-based study was carried out between February and March 2021 in Germany and Czech Republic among healthcare workers who recently received the AstraZeneca COVID-19 vaccine. The study used a validated self-administered questionnaire composed of twenty-eight multiple-choice items covering demographic variables, medical anamneses, and local, systemic, oral, and skin related SEs of the vaccine. Out of the ninety-two included participants, 77.2% were females and 79.3% were from Germany. Their mean age was 35.37 ± 12.62 (19–64) years-old, 15.2% had chronic illnesses and 22.8% were receiving medical treatments. Overall, 94.6% of the participants reported at least one SE. The most common local SE was injection site pain (72.8%), and the most common systemic SEs were fatigue (73.9%), muscle pain (55.4%), chills (48.9%), feeling unwell (46.7%), nausea (45.7%), and headache (29.3%). The vast majority (91.9%) resolved within 1–3 days, and the below 35 years-old group was the least affected age group. The SEs’ frequency was insignificantly higher in females and previously infected participants; the vaccine safety for the elderly was supported by the early findings of this study. Chronic illnesses and medical treatments were not associated with an increased risk of SE incidence and frequency. No blood disorder SEs were reported in our sample. Further independent studies are highly required to evaluate the safety of the AstraZeneca vaccine and to explore whether gender or previous infection could be associated with the vaccine SEs.

## 1. Introduction

Since the beginning of the coronavirus pandemic, researchers have been working tirelessly to find a solution to prevent the virus spread. Developing an effective vaccine was the most eminent priority for such a public health disaster [1]. The Oxford–AstraZeneca COVID-19 vaccine (ChAdOx1 nCoV-19 vaccine/AZD1222) is one of the developed viral vector vaccines with an efficacy of 70.4% [2]. It is produced from an adenovirus that normally infects chimpanzees. The adenovirus contains the gene for the synthesis of the complete spike protein of a coronavirus. The spike proteins synthesized in the vaccinated person’s body trigger an immune response against infection with SARS-CoV-2 [3,4]. Unlike the mRNA vaccines, adenovirus vector vaccines do not require ultra-low storage temperature and, therefore, could be feasiblly distributed and delivered [5].

The first approval for the Oxford–AstraZeneca COVID-19 vaccine was an emergency approval granted in the United Kingdom on 30th December 2020. Authorization in the European Union was granted on 29th January 2021 by the European Medicines Agency (EMA, Amsterdam, The Netherlands) [6].

The AstraZeneca COVID-19 vaccination course consists of two separate doses of 0.5 mL each. The second dose should be administered between 4 and 12 weeks (28 to 84 days) after the first dose [7,8]. The frequently reported side effects (SEs) were injection site tenderness, injection site pain, headache, fatigue, myalgia, malaise, pyrexia, chills, arthralgia, and nausea [9]. The majority of them were mild to moderate and usually resolved within a few days of vaccination [9]. When compared with the first dose, SEs after the second dose were milder and reported less frequently [9].

While oral and skin-related SEs after vaccination are either missing or underestimated, COVID-19 was frequently associated with a wide array of mucocutaneous SEs that emerged intra-orally and extra-orally [10,11,12,13,14,15,16,17]. Moreover, several vaccines are associated with oral SEs, including hepatitis B, diphtheria, and tetanus vaccine [18,19].

Many European Union (EU, Brussels, Belgium) states recently suspended vaccinations using the AstraZeneca COVID-19 vaccine due to individual reports of cerebral sinus thrombosis and mesenteric vein thrombosis [20,21,22]. The EMA safety committee reported in an extraordinary meeting of 18th March 2021 the incidence of thrombosis and thrombocytopenia in persons who had recently received AstraZeneca COVID-19 vaccine, mostly within 14 days after vaccination [23]. The majority of reports involved women under 55 years-old [23].

Lack of independent studies on vaccines’ safety may adversely impact the vaccine uptake, which has to be accelerated in the following months. Therefore, this study’s primary objective was to estimate the prevalence of AstraZeneca COVID-19 vaccine SEs among the early vaccinated healthcare workers in Germany and Czech Republic. The secondary objective was to explore the potential risk factors of SEs’ occurrence.

## 2. Materials and Methods

### 2.1. Study Design

A multi-center cross-sectional study was carried out from 5th February to 16th March 2021 to evaluate the short-term side-effects of COVID-19 vaccines, including the AstraZeneca vaccine, in Germany and Czech Republic. An online self-administered questionnaire was developed using KoBoToolbox (Harvard Humanitarian Initiative, Cambridge, MA, USA) to collect data from both countries’ target participants [24]. The study had been registered in clinicaltrials.gov with an identifier number NCT04706156 (Oral Side Effects of COVID-19 vaccine), and it was reported according to the STROBE guidelines for cross-sectional studies [25,26].

### 2.2. Participants

The healthcare workers who received the AstraZeneca COVID-19 vaccine during the early vaccination phase as priority groups (high-risk groups) were the target population of this study. Non-healthcare workers who received the AstraZeneca COVID-19 vaccine and healthcare workers who received the vaccine earlier than 29th January 2021 through phase 3 trials were excluded.

The participants were invited to this study through printed posters in the vaccination center (Kiel, Germany) and distribution lists of the Czech Clinical Research Infrastructure Network (CZECRIN) and the Institute of Health Information and Statistics of the Czech Republic (IHIS-CR, Prague, Czech Republic) [27,28]. The participants received no financial compensation, and they were offered to withdraw from the study at any moment before submitting their answers.

### 2.3. Instrument

A self-administered questionnaire with 28 multiple-choice items was adapted from previous studies on influenza vaccine SEs and the authorization reports of the Centers for Disease Control and Prevention (CDC, Atlanta, GA, USA) for COVID-19 vaccines [18,29]. The whole process of adaptation, validation and reliability testing of the questionnaire was described in detail previously [25,30,31]. Experts panel validation and test re-test reliability were utilized to validate the suggested instrument with a mean Cohen’s kappa coefficient of 0.89 ± 0.13 (0.54–1). Dual forward translation and bi-lingual experts panel were employed to produce German and Czech versions of the instrument.

The questionnaire had four basic categories: (a) demographic data including gender, age, profession, length of work experience, and region; (b) medical anamneses including chronic illnesses and medical treatments; (c) COVID-19-related anamneses including previous infection, exposure, type of vaccine, and the number of doses; (d) the systemic, local, oral, and skin-related SEs of COVID-19 vaccine.

### 2.4. Ethical Considerations

The study protocol was reviewed and approved by the Ethics Committee of the Faculty of Medicine at Masaryk University on 20th January 2021 (Ref. 2/2021) and Faculty of Medicine at Justus Liebig University of Giessen (Ref. 55/20). Masaryk University was the data controller, and data acquisition and processing followed the General Data Protection Regulation (GDPR) [32]. Identifying personal data were not collected from participants, and the participants had to provide their informed consent digitally before joining the study.

### 2.5. Statistical Analysis

The Statistical Package for the Social Sciences (SPSS) version 27.0 (SPSS Inc. Chicago, IL, USA, 2020) was used to carry out all statistical tests [33]. Descriptive statistics were executed for the demographic data, medical anamnesis, COVID-19-related anamnesis and the SEs of the AstraZeneca COVID-19 vaccine represented by frequencies, percentages, cumulative percentages, means and standard deviation. Consequently, inferential statistics were performed to explore the potential risk factors of the SEs. Chi-squared test (*χ*^2^), Fisher’s exact test, and Mann–Whitney U test (*U*) were used with a confidence level of 95% and a significance value (*Sig.*) ≤ 0.05. Moreover, binary logistic regression was conducted to calculate the odds ratio of local, systemic, and general SE occurrence with a confidence level of 95% and a significance value (*Sig.*) ≤ 0.05.

## 3. Results

### 3.1. Demographic Characteristics

A total of ninety-two participants were included in this study, while eight were excluded because they received the vaccine earlier than the mass rollout of AstraZeneca COVID-19 vaccine in the EU. Seventy-one (77.2%) participants were females, and twenty-one (22.8%) were males. Participants’ mean age was 35.37 ± 12.62 (19–64) years-old, and 19.6% of them were above 49 years old. Age stratification was carried out to facilitate subsequent analyses where the median age (35 years-old) and the predicted age limit of thrombotic events (50 years-old) were used as cutoff points [20]. Twelve (13%) participants were physicians, ten (10.9%) were dentists, twenty-six (28.3%) were nurses, and the rest were allied health professionals. The majority (45.7%) of participants had 1–5 years of working experience, and 27% had more than 20 years of experience. In Germany, the majority (68.5%) of participants were from Schleswig-Holstein state, while in Czech Republic, the majority (63.2%) were from Vysočina region (Table 1).

### 3.2. Medical Anamneses

A total of fourteen (15.2%) participants reported suffering from at least one chronic illness. The most prevalent was thyroid disease (5.4%), followed by asthma (4.3%) and neurologic disease (3.3%). The above 49 years-old group was the most affected (22.2%) without any statistically significant difference between the age groups, and females were more affected than males without statistical significance (*χ*^2^ = 0.684; *p* = 0.408), 16.9% vs. 9.5%, respectively (Table 2).

A total of twenty-one (22.8%) participants reporting taking medical treatments regularly. The most prevalent drugs were antidepressants (8.7%), followed by thyroid hormone supplement (6.5%) and contraceptives (4.3%). The 35–49 years-old age group was the most frequent drug consuming (30.4%), and females were more consuming than males without statistical significance (*χ*^2^ = 1.127; *p* = 0.288), 25.4% vs. 14.3%, respectively (Table 2).

### 3.3. COVID-19-Related Anamneses

A total of eight (8.7%) participants were previously infected by SARS-CoV-2, and the 35–49 years-old group was the most affected (13%). On the other hand, thirty-five (38%) participants reported being exposed to COVID-19 patients in the previous months; also, the 35–49 years-old group was the most exposed (47.8%).

Across genders, males and females were equally exposed to COVID-19 patients, 38.1% and 38%, respectively; males were more infected by SARS-CoV-2 than females without statistically significant difference (*χ*^2^ = 1.071; *p* = 0.301), 14.3% vs. 7%, respectively. All the included participants received one dose of AstraZeneca COVID-19 vaccine, except one (1.1%) participant from Usti nad Labem, Czech Republic (Table 2).

### 3.4. Local and Systemic Side Effects

A total of eighty-seven (94.6%) participants reported having at least one local or systemic SE after receiving AstraZeneca COVID-19 vaccine. The local SEs were experienced by sixty-eight (73.9%) participants, and the most affected age group was the below 35 years-old group (82.4%), followed by the 35–49 years-old group (73.9%), and the above 49 years-old group (50%). The most common local SE was injection site (72.8%), while injection site swelling, and injection site redness were experienced by 10.9% of the participants for each of them. Injection site redness was more common among the 35–49 years-old group (17.4%) than other age groups.

The systemic SEs were experienced by seventy-nine (85.9%) participants, and it was more common among the below 35 years-old group (96.1%) than other groups. The most common systemic SE was fatigue (73.9%), followed by muscle pain (55.4%), chills (48.9%), feeling unwell (46.7%), nausea (45.7%), and headache (29.3%).

In all reported SEs, the below 35 years-old group was the most affected and the above 49 years-old group was the least affected. This pattern was not existing in injection site redness and in fever, where the 35–49 years-old group was the most affected, and the above 49 years-old group was still the least affected. The difference among age groups in terms of local SEs prevalence and systemic SEs prevalence was statistically significant (*χ*^2^ = 7.222 and 9.853; *p* = 0.027 and 0.007, respectively).

Out of the ninety-two participants of this study, two female participants (20 years-old and 28 years-old) from Germany reported experiencing severe SEs that made them seek medical care.

The vast majority of SEs remained either for one day (56.3%) or three days (35.6%) postvaccination; thus, indicating that 91.1% of the SEs would resolve within three days after vaccination. The 35–49 years-old group was the most affected by longer-standing SEs, 9.5% for five days and 9.5% for one week. The mean total number of SEs was 4.57 ± 2.71, and it was significantly different (*H* = 13.006; *p* = 0.001) across the age groups. The mean of the below 35 years-old group was 5.35 ± 2.40, and the mean of the above 49 years-old group was 2.89 ± 2.45 (Table 3).

### 3.5. Oral and Skin-Related Side Effects

Mouth sores including vesicles, blisters and ulcers were the most commonly reported oral SEs (7.6%), followed by bleeding taste alterations (5.4%), and bleeding gingiva (3.3%). It is worthy to mention that taste alterations were unsolicited SEs, and they were mainly metallic-like. The onset of oral SEs was mainly 1–3 days (82.6%), and the most common locations were tongue, labial or buccal mucosa, and lips. The mean total number of oral SEs was 0.22 ± 0.51 (0–5), and the highest mean was in the below 35 years-old group while the lowest mean was in the above 49 years-old group. Skin rash was the only skin-related SE reported by the participants (4.3%); two participants were below 35 years-old, and one participant was above 49 years-old (Table 4).

### 3.6. Risk Factors of AstraZeneca COVID-19 Vaccine Side Effects

In terms of local SEs, the mean total was higher in females (0.97 ± 0.77) than males (0.86 ± 0.73), in the ≤50 years-old group (1.03 ± 0.73) than the >50 years-old group (0.56 ± 0.81), and in the previously infected participants (1.38 ± 0.92) than the non-infected participants (0.90 ± 0.74). The same pattern was found for the systemic SEs, as female participants had an insignificantly higher mean total (3.72 ± 2.34) than the males (3.29 ± 2.22), also the ≤ 50 years-old group (3.91 ± 2.30) was higher than the >50 years-old group (2.25 ± 1.88), and the previously infected participants experienced insignificantly more systemic SEs (4.38 ± 2.13) than the non-infected participants (3.55 ± 2.33).

In general, the female gender (*U* = 680; *p* = 0.539), the younger age group (*U* = 322.5; *p* = 0.003), and the previous COVID-19 infection group (*U* = 232; *p* = 0.149) were associated with more SEs following AstraZeneca COVID-19 vaccine.

Chronic illnesses were associated with slightly less local SEs and considerably less systemic SEs. The same pattern was found for medical treatments, as the participants who received medical treatments experienced slightly less local SEs and considerably less systemic SEs (Table 5).

The binary logistic regression revealed that females had an odds ratio (OR) to experience SEs after receiving AstraZeneca COVID-19 vaccine of 5.750 times (CI 95%: 0.893–37.043) higher than their male counterparts. The younger age group (≤50 years-old) had an OR of 3.476 times (CI 95%: 0.531–22.744) higher than the older age group to experience SEs generally. Prior COVID-19 infection was associated with an increased OR of 2.639 (CI 95%: 0.308–22.646) for the local SEs, and ab OR of 1.167 (CI 95%: 0.132–10.347) for the systemic SE (Table 6).

On the contrary, the participants with chronic illnesses had an OR of 0.703 times (CI 95%: 0.073–6.796), and the participants taking medical treatments had an OR of 0.174 times (CI 95%: 0.027–1.120) less than their counterparts to experience SEs after the AstraZeneca COVID-19 vaccine (Table 6).

## 4. Discussion

Unquestionably, the main pathway to overcome the economic and public health burdens of COVID-19 pandemic is through accelerating “mass vaccination” efforts [34,35,36,37,38]. The beginning of the vaccination split the pandemic history to before- and after-vaccine and gave a hope to the globe that pandemic-induced restrictions and limitations will be lifted [1,39,40,41]. The Oxford–AstraZeneca (ChAdOx1 nCoV-19) vaccine acquired emergency approval by the UK Medicines and Healthcare products Regulatory Agency (MHRA, London, UK) in December 2020, and it was the third approved vaccine in the EU by the EMA in late January 2021 [42,43,44].

In the EMA product information report of the AstraZeneca COVID-19 vaccine, the overall safety was based on pooled data of 23,745 participants from the United Kingdom (UK), Brazil and South Africa. The data after the first and second dose were not analyzed separately (about half were vaccinated with one dose), and the time after the vaccination is underreported as well in the EMA report [9]. The injection site tiredness was the most commonly reported local SE (63.7%) in the EMA report, although it was not solicited in this study’s questionnaire. Injection site pain was found more frequently in our European sample (72.8%) compared to the EMA report (54.2%). Moreover, injection site swelling (10.9%) and injection site redness (10.9%) were solicited in our study; however, they were not reported by the manufacturer. Compared with the CDC report on Pfizer-BioNTech COVID-19 vaccine, the injection site redness and swelling were slightly more prevalent among our participants who received the AstraZeneca vaccine [29].

In terms of systemic SEs, headache (52.6%) was the most commonly reported SE by the manufacturer; however, it was considerably less prevalent in our sample (29.3%). Contrarily, fatigue, nausea, muscle pain, and joint pain were more prevalent in our sample (73.9%, 45.7%, 55.4%, and 41.3%, respectively) than in the EMA report (53.1%, 21.9%, 44%, and 26.4%, respectively). However, being described as an “uncommon” SE by the manufacturer, lymphadenopathy was reported by 5.4% of our sample.

The second dose was associated with lower frequency and intensity of the reported SEs by the manufacturer, and this finding was not examined by our study as 98.9% of the participants received only the first dose by the time of filling in the questionnaire [9]. The EMA report claimed that the majority of SEs would resolve within a few days, and our findings support this finding as 91.9% of the SEs resolved within three days, and the rest resolved within the first week [9].

The vaccine safety for the elderly was suggested by the EMA report, as reactogenicity was milder and the SEs were less frequently reported by older adults. The results of our study explicitly support this suggestion, as the frequency of SEs was correlated inversely with age (*ρ* = −0.386; *p* < 0.001). The people above 50 years-old experienced significantly (*U* = 322.5; *p* = 0.003) less frequent SEs. It is important to take into consideration that the German Standing Committee on Vaccination (STIKO, Berlin, Germany) limited the vaccination using AstraZeneca COVID-19 vaccine to people between 18 and 64 years-old [44].

The manufacturer did not report the gender-related differences except for blood disorders, including clots and thrombocytopenia, which were rarely reported by female cases (under 55 years-old). However, female participants in our study experienced more frequent local and systemic SEs without statistical significance, and the small sample size of our study suggests that those difference should be further investigated in large-scale independent studies. On the other hand, none of the blood disorder SEs were reported in our sample.

Previous infection was associated with more frequent SEs according to our study’s data; therefore, future independent studies should explore the role of previous COVID-19 infection and residual antibodies in the frequency and intensity of SEs. The patency period between the recovery date and the first vaccination dose should be taken into consideration. In contrast to COVID-19, chronic illnesses and medical treatments were not associated with an increased risk of AstraZeneca COVID-19 vaccine SEs. The SEs were less frequent among participants with chronic illnesses and medical treatments. This preliminary finding has to be examined in population-based studies.

Regarding the incidence of blood disorders following AstraZeneca COVID-19 vaccine, thirty-seven reported cases of thrombosis led to suspension of vaccination in sixteen EU states and immediate investigation by the EMA [20]. Within the limits of our study, no blood disorder SEs were reported; however, the EMA reported thrombosis in several blood vessels of seven cases and eighteen people had cerebral venous sinus thrombosis until 17th March 2021 [20]. Taking into consideration that more than eighteen million adults received the AstraZeneca COVID-19 vaccine (eleven million in the UK, and seven million in the EU), and the fact that 100,000 blood clots are recorded monthly in the EU and another 3000 clots in the UK, these blood disorders following the vaccine were deemed as rare side effects [20]. In this context, an online conference held on 19th March 2021, by a scientific group from the University of Greifswald (Greifswald, Germany) in collaboration with Paul Ehrlich Institute (Langen, Germany), proposed that there might be a link between the rarely reported blood clots and the vaccine. The group leader Prof. Andreas Greinacher claimed that antibodies, which in rare cases formed after vaccination, activated the blood platelets. These then acted similar to wound healing and triggered thrombosis in the brain [45]. This should be treated as a potential hypothetical explanation for the extremely rare SEs, and further information was promised to be revealed by Prof. Greinacher’s group shortly [46].

This study’s first limitation is its small sample size; however, this should be viewed as initial evidence that was required to be exposed urgently. As the AstraZeneca COVID-19 vaccine was approved by EMA later than Pfizer-BioNTech and Moderna vaccines, it is a less prevalent vaccine among healthcare workers, which is another reason for the smaller sample size. The second limitation is related to the nature of phase IV trials based on the subjective self-reported and self-assessed outcomes; therefore, healthcare workers were targeted in this study as they have high levels of health literacy and scientific interest. Moreover, Jęśkowiak et al. in 2021 found that the outcomes of governmental passive surveillance systems are unreliable to estimate the prevalence of COVID-19 vaccines’ SEs, thus suggesting that the post-marketing (phase IV) studies that utilize self-reported outcomes could be the best available method to achieve this purpose [47]. The third limitation is that the instrument of this study did not inquire about the details of severe SEs that required medical care nor the timing of side effects in relation to the first or second shot.

To the best of the authors’ knowledge, this report provides the first independent evidence on AstraZeneca COVID-19 vaccine short-term SEs. Further studies on this vaccine’s safety and SEs in the short- and long-term are deemed required for ensuring public confidence in vaccines.

## 5. Conclusions

The prevalence of AstraZeneca COVID-19 vaccine SEs seems to be more frequent among young age groups, thus suggesting higher vaccine safety levels for the elderly. Within this study’s limits, the prevalence of AstraZeneca COVID-19 vaccine SEs was insignificantly higher in the previously infected participants and females; however, none of the blood disorder SEs were reported in our sample. Chronic illnesses and medical treatments were not associated with an increased risk of SEs’ occurrence and frequency. Further independent studies are highly required to evaluate the vaccine’s overall safety and estimate more precisely the prevalence of each local and systemic SE.

## Figures and Tables

**Table 1 vaccines-09-00673-t001:** Demographic characteristics of European healthcare workers receiving AstraZeneca COVID-19 vaccine, February–March 2021.

Variable			Frequency	Percentage	Cumulative Percentage
**Gender**	Female		71	77.2%	77.2
	Male		21	22.8%	100%
**Age**	<35 years-old		51	55.4%	55.4%
	35–49 years-old		23	25%	80.4%
	>49 years-old		18	19.6%	100%
**Profession**	Physician		12	13%	13%
	Dentist		10	10.9%	23.9%
	Nurse		26	28.3%	52.2%
	Other allied Health Professional		44	47.8%	100%
**Experience**	1–5 years		42	45.7%	45.7%
	6–10 years		12	13%	58.7%
	11–20 years		13	14.1%	72.8%
	>20 years		25	27.2%	100%
**Region**	**Germany**	Schleswig-Holstein	50	54.3%	54.3%
		Bayern	19	20.7%	75%
		Nordrhein-Westfalen	3	3.3%	78.3%
		Hessen	1	1.1%	79.3%
	**Czech Republic**	Praha	1	1.1%	80.4%
		Vysočina	11	12%	92.4%
		Hradec Kralove	3	3.3%	95.7%
		Moravian-Silesian	2	2.2%	97.8%
		Central Bohemian	1	1.1%	98.9%
		Usti nad Labem	1	1.1%	100%

**Table 2 vaccines-09-00673-t002:** Medical anamneses of European healthcare workers receiving the AstraZeneca COVID-19 vaccine, February–March 2021.

Variable			<35 Years-Old	35–49 Years-Old	>49 Years-Old	Total	*Sig.* ^1^
**Chronic Illness**	Asthma		3 (5.9%)	0 (0%)	1 (5.6%)	4 (4.3%)	0.640
	Allergy		0 (0%)	0 (0%)	1 (5.6%)	1 (1.1%)	0.196
	Bone Disease		1 (2%)	0 (0%)	0 (0%)	1 (1.1%)	1.000
	Chronic Hypertension		0 (0%)	0 (0%)	1 (5.6%)	1 (1.1%)	0.196
	Neurologic Disease		0 (0%)	1 (4.3%)	2 (11.1%)	3 (3.3%)	0.049
	Psychologic Distress		0 (0%)	1 (4.3%)	1 (5.6%)	2 (2.2%)	0.196
	Rheumatoid Arthritis		1 (2%)	0 (0%)	0 (0%)	1 (1.1%)	1.000
	Thyroid Disease		3 (5.9%)	2 (8.7%)	0 (0%)	5 (5.4%)	0.590
	Total		6 (11.8%)	4 (17.4%)	4 (22.2%)	14 (15.2%)	0.462
**Medical Treatment**	Antidepressants		3 (5.9%)	2 (8.7%)	3 (16.7%)	8 (8.7%)	0.367
	Antiepileptics		0 (0%)	1 (4.3%)	0 (0%)	1 (1.1%)	0.446
	Antihypertensive		0 (0%)	1 (4.3%)	0 (0%)	1 (1.1%)	0.446
	Contraceptives		3 (5.9%)	0 (0%)	1 (5.6%)	4 (4.3%)	0.640
	Immunosuppressive		1 (2%)	0 (0%)	0 (0%)	1 (1.1%)	1.000
	Pain killers		0 (0%)	0 (0%)	1 (5.6%)	1 (1.1%)	0.196
	Thyroid Hormone		3 (5.9%)	3 (13%)	0 (0%)	6 (6.5%)	0.216
	Total		10 (19.6%)	7 (30.4%)	4 (22.2%)	21 (22.8%)	0.626
**COVID-19 anamneses**	Previous Infection		4 (7.8%)	3 (13%)	1 (5.6%)	8 (8.7%)	0.692
	Exposure		19 (37.3%)	11 (47.8%)	5 (27.8%)	35 (38%)	0.416
	**Vaccine Dosage**	One Dose	51 (56%)	22 (24.2%)	18 (19.8%)	91 (98.9%)	0.446
		Two Doses	0 (0%)	1 (100%)	0 (0%)	1 (1.1%)	

^1^ Chi-squared test and Fisher’s exact test were used with a significance level (*Sig.*) of <0.05.

**Table 3 vaccines-09-00673-t003:** Local and systemic side effects of AstraZeneca COVID-19 vaccine among European healthcare workers, February–March 2021.

Variable	Outcome	<35 Years-Old	35–49 Years-Old	>49 Years-Old	Total	*Sig.* ^1^
**Local SEs**	Injection Site Pain	42 (82.4%)	16 (69.6%)	9 (50%)	67 (72.8%)	0.028
	Injection Site Swelling	8 (15.7%)	1 (4.3%)	1 (5.6%)	10 (10.9%)	0.343
	Injection Site Redness	5 (9.7%)	4 (17.4%)	1 (5.6%)	10 (10.9%)	0.539
	Total	42 (82.4%)	17 (73.9%)	9 (50%)	68 (73.9%)	0.035
**Systemic SEs**	Fatigue	46 (90.2%)	13 (56.5%)	9 (50%)	68 (73.9%)	<0.001
	Headache	18 (35.3%)	8 (34.8%)	1 (5.6%)	27 (29.3%)	0.040
	Nausea	28 (54.9%)	8 (34.8%)	6 (33.3%)	42 (45.7%)	0.138
	Feeling Unwell	26 (51%)	10 (43.5%)	7 (38.9%)	43 (46.7%)	0.634
	Muscle Pain	33 (64.7%)	12 (52.2%)	6 (33.3%)	51 (55.4%)	0.066
	Joint Pain	24 (47.1%)	8 (34.8%)	6 (33.3%)	38 (41.3%)	0.456
	Fever	8 (15.7%)	5 (21.7%)	1 (5.6%)	14 (15.2%)	0.353
	Chills	31 (60.8%)	9 (39.1%)	5 (27.8%)	45 (48.9%)	0.031
	Lymphadenopathy	4 (7.8%)	1 (4.3%)	0 (0%)	5 (5.4%)	0.707
	Total	49 (96.1%)	17 (73.9%)	13 (72.2%)	79 (85.9%)	0.003
**SEs** **Duration**	1 day	27 (54%)	10 (47.6%)	12 (75%)	49 (56.3%)	0.136
	3 days	20 (40%)	7 (33.3%)	4 (25%)	31 (35.6%)	
	5 days	2 (4%)	2 (9.5%)	0 (0%)	4 (4.6%)	
	1 week	1 (2%)	2 (9.5%)	0 (0%)	3 (3.4%)	
	Total	50 (98%)	21 (91.3%)	16 (88.9%)	87 (94.6%)	0.141
**Total SEs**	(0–12)	5.35 ± 2.40	4.13 ± 2.94	2.89 ± 2.45	4.57 ± 2.71	0.001

^1^ Chi-squared test, Fisher’s exact test and Kruskal–Wallis test were used with a significance level of <0.05.

**Table 4 vaccines-09-00673-t004:** Oral and skin-related side effects of AstraZeneca COVID-19 vaccine among European healthcare workers, February–March 2021.

Variable	Outcome	<35 Years-Old	35–49 Years-Old	>49 Years-Old	Total	*Sig.* ^1^
**Solicited Oral SE**	Ulcers/Blisters/Vesicles	4 (7.8%)	3 (13%)	0 (0%)	7 (7.6%)	0.423
	White/Red Plaque	1 (2%)	0 (0%)	0 (0%)	1 (1.1%)	1.000
	Halitosis	1 (2%)	0 (0%)	2 (11.1%)	3 (3.3%)	0.147
	Bleeding Gingiva	2 (3.9%)	1 (4.3%)	0 (0%)	3 (3.3%)	1.000
	Swollen Lips	1 (2%)	0 (0%)	0 (0%)	1 (1.1%)	1.000
**Unsolicited Oral SE**	Taste alterations	3 (5.9%)	1 (4.3%)	1 (5.6%)	5 (5.4%)	1.000
**Onset**	1–3 days	12 (100%)	3 (42.9%)	4 (100%)	19 (82.6%)	0.005
	Within 1st week	0 (0%)	3 (42.9%)	0 (0%)	3 (13%)	
	Within 4th week	0 (0%)	1 (14.3%)	0 (0%)	1 (4.3%)	
	Total	9 (17.6%)	5 (21.7%)	2 (11.1%)	16 (17.4%)	0.654
**Oral SE Location**	Lips	0 (0%)	1 (4.3%)	0 (0%)	1 (1.1%)	0.446
	Labial/Buccal Mucosa	2 (3.9%)	2 (8.7%)	0 (0%)	4 (4.3%)	0.506
	Tongue	2 (3.9%)	0 (0%)	0 (0%)	2 (2.2%)	1.000
**Total Oral SE**	(0–5)	0.24 ± 0.55	0.22 ± 0.42	0.17 ± 0.51	0.22 ± 0.51	0.739
**Skin-related SE**	Skin Rash	2 (3.9%)	1 (4.3%)	1 (5.6%)	4 (4.3%)	1.000

^1^ Chi-squared test, Fisher’s exact test and Kruskal–Wallis test were used with a significance level of <0.05.

**Table 5 vaccines-09-00673-t005:** Risk factors of AstraZeneca COVID-19 vaccine side effects among European healthcare workers, February–March 2021.

Variable	Outcome	Local SE (*n*)	Systemic SE (*n*)	Total	*Sig.* ^1^
**Gender**	Female	0.97 ± 0.77	3.72 ± 2.34	4.69 ± 2.74	0.539
	Male	0.86 ± 0.73	3.29 ± 2.22	4.14 ± 2.61	
**Age Group**	≤50 years-old	1.03 ± 0.73	3.91 ± 2.30	4.93 ± 2.63	0.003
	>50 years-old	0.56 ± 0.81	2.25 ± 1.88	2.81 ± 2.43	
**Infection**	No	0.90 ± 0.74	3.55 ± 2.33	4.45 ± 2.70	0.149
	Yes	1.38 ± 0.92	4.38 ± 2.13	5.75 ± 2.61	
**Chronic Illness**	No	0.95 ± 0.75	3.73 ± 2.27	4.68 ± 2.61	0.230
	Yes	0.93 ± 0.83	3.00 ± 2.54	3.93 ± 3.22	
**Medical Treatment**	No	0.96 ± 0.71	3.73 ± 2.20	4.69 ± 2.53	0.294
	Yes	0.90 ± 0.94	3.24 ± 2.66	4.14 ± 3.28	

^1^ Mann–Whitney test was used with a significance level of <0.05.

**Table 6 vaccines-09-00673-t006:** Regression analysis of AstraZeneca COVID-19 vaccine side effects’ risk factors, February–March 2021.

Predictor		Local SE		Systemic SE		Total SE	
		**Odds Ratio**	***Sig***.	**Odds Ratio**	***Sig***.	**Odds Ratio**	***Sig***.
**Gender**	Female (vs. Male)	1.588 (0.551–4.577)	0.392	2.461 (0.709–8.545)	0.156	5.750 (0.893–37.043)	0.066
**Age Group**	≤50 years (vs. >50 years)	5.229 (1.676–16.313)	0.004	2.481 (0.657–9.368)	0.180	3.476 (0.531–22.744)	0.194
**Infection**	Yes (vs. No)	2.639 (0.308–22.646)	0.376	1.167 (0.132–10.347)	0.400		
**Chronic Illness**	Yes (vs. No)	0.862 (0.243–3.058)	0.818	0.539 (0.128–2.273)	0.890	0.703 (0.073–6.796)	0.761
**Medical Treatment**	Yes (vs. No)	0.473 (0.167–1.340)	0.159	0.406 (0.117–1.411)	0.156	0.174 (0.027–1.120)	0.066

Logistic regression analysis was performed with a significance level (*Sig.*) of <0.05 and confidence level (CI) of 95%.

## Data Availability

The data that support the findings of this study are available from the corresponding author upon reasonable request.

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
