# Peer review of "Safety of ChAdOx1 nCoV-19 Vaccine: Independent Evidence from Two EU States"

_vaccines, 2021, doi:10.3390/vaccines9060673_

Round 1

Reviewer 1 Report

Independent studies like this are absolutely needed, but the data set is too small to support the claims.

Author Response

Dear Reviewer,

We are delighted to have the opportunity to revise and resubmit our manuscript titled “Safety of ChAdOx1 nCoV-19 Vaccine: Independent Evidence from Two EU States” (Manuscript ID vaccines-1237179).

We have considered all remarks provided by all of the reviewers. Please find appended a revised version of the manuscript (with track changes highlighted) and a point-by-point rebuttal to all comments raised as detailed below. We hope our responses are satisfactory in addressing the criticisms and suggestions.

We hope the revised manuscript will be in an acceptable format. Thank you for your kind contribution.

  1. Independent studies like this are absolutely needed, but the data set is too small to support the claims.

Answer:

We thank the reviewer for giving us the opportunity to explain this point.

A) We agree that our study would have benefited from having a larger sample; however, we also believe that every single piece of independent evidence (non-sponsored studies) on the vaccines' safety will benefit the public health systems and will increase public confidence in vaccines.

B) In both Germany and the Czech Republic, the AstraZeneca vaccine was not given primarily to healthcare workers (our target population); therefore, we did our best to reach healthcare workers who recieved this vaccine but it was not easy to have a larger sample.

C) It was not also possible to have a larger sample at the time of writing the first version of the manuscript in March when the first thrombotic disorders were reported in relation to the AstraZeneca vaccine. As we aimed to disseminate our findings as soon as possible to benefit the health policies makers.

Sincerely,

Reviewer 2 Report

Estimated Authors,

Estimated Editors,

I've read with great interest the paper from the study group lead by Riad A and reporting on the safety profile of ChAdOx1 vaccine from Germany and Czech Republic.

The paper is of certain interest, as it deals with a significant public Health Problem, and being based in two different EU countries  intrinsically increases the potential relevance of its content.

However, in my opinion, the present paper is affected by several shortcomings that impair - at least temporarily, the eventual acceptance of this otherwise interesting research work.

The main one is represented by the very small sample size. This problem is properly addressed by the Authors in the discussion, but some further comments on the potential significance, reliability and generalizability of their results would be welcome. According with the outcome inquired by the study researchers, they should report the hypotethical minimal sample size required, and discuss accordingly how the lack of the target sample impaired the overall generalizability of their results.

Second, while data reporting was (at least in my opinion) accurate, data analysis should be improved in order to increase the overall significance of the paper.

For example, I would suggest to perform a logistic regression having as the outcome variable the reporting of:

1) any SE

2) any local SE

3) any systemic SE

and as explanatory variables those reported in Table 1+2

Some minor remarks:

In table 1 and Table 2, Authors are not actually reporting the "outcome", but rather the variables, please fix accordingly (on the contrary, the use of the term "outcome" is correct in Table 3 and 4).

Finally, when dealing with the limitation of discussing self-reported signs/symptoms, Authors should provide some references on the reliability of similar studies.

Author Response

Dear Reviewer,

We are delighted to have the opportunity to revise and resubmit our manuscript titled “Safety of ChAdOx1 nCoV-19 Vaccine: Independent Evidence from Two EU States” (Manuscript ID vaccines-1237179).

We have considered all remarks provided by all of the reviewers. Please find appended a revised version of the manuscript (with track changes highlighted) and a point-by-point rebuttal to all comments raised as detailed below. We hope our responses are satisfactory in addressing the criticisms and suggestions.

We hope the revised manuscript will be in an acceptable format. Thank you for your kind contribution.

  1. The main one is represented by the very small sample size. This problem is properly addressed by the Authors in the discussion, but some further comments on the potential significance, reliability and generalizability of their results would be welcome. According with the outcome inquired by the study researchers, they should report the hypotethical minimal sample size required, and discuss accordingly how the lack of the target sample impaired the overall generalizability of their results.

Answer:

We thank the reviewer for giving us the opportunity to explain this point.

A) We agree that our study would have benefited from having a larger sample; however, we also believe that every single piece of independent evidence (non-sponsored studies) on the vaccines' safety will benefit the public health systems and will increase public confidence in vaccines.

B) In both Germany and the Czech Republic, the AstraZeneca vaccine was not given primarily to healthcare workers (our target population); therefore, we did our best to reach healthcare workers who received this vaccine but it was not easy to have a larger sample.

C) It was not also possible to have a larger sample at the time of writing the first version of the manuscript in March when the first thrombotic disorders were reported in relation to the AstraZeneca vaccine. As we aimed to disseminate our findings as soon as possible to benefit the health policies makers.

D) The optimal sample size was supposed to be ≃ 386 assuming that the expected frequency of the outcome is 50%, CI 95%, and error margin 5%. However, due to the reasons mentioned above, it was not feasible to reach this optimal size; therefore, we tried in our analysis to give more focus on the descriptive analysis rather than the inferential statisitics because we believe that our sample was small.

2. Second, while data reporting was (at least in my opinion) accurate, data analysis should be improved in order to increase the overall significance of the paper. For example, I would suggest to perform a logistic regression having as the outcome variable the reporting of: 1) any SE; 2) any local SE; 3) any systemic SE. and as explanatory variables those reported in Table 1+2.

Answer:

We sincerely thank the reviewer for this suggestion and we did implement in the results as well as the methods section. Line 135 - 137, Line 240 - 250 & Table 6

3. In table 1 and Table 2, Authors are not actually reporting the "outcome", but rather the variables, please fix accordingly (on the contrary, the use of the term "outcome" is correct in Table 3 and 4).

Answer:

We agree with this point and implemented the suggested changes. Table 1 &2

4. Finally, when dealing with the limitation of discussing self-reported signs/symptoms, Authors should provide some references on the reliability of similar studies.

Answer:

We thank the reviewer for giving us the opportunity to reflect more on this point. In general, post-marketing (phase IV) studies of drug safety rely mainly on the self-reported outcomes; therefore, our study used self-administered questionnaire to facilitate data collection. The most recent evidence from Poland revealed that the passive surveillance systems are unreliable to determine the prevalence of postvaccination SE in comparison of the active survellience (phase IV) studies. https://doi.org/10.3390/vaccines9050502

We added the suggested point to the Discussion section. Line 333 - 336

Sincerely,

Reviewer 3 Report

I read with interest this paper that, despite the low sample size, represent an important work.

Regarding the validation of the questionnaire few details were reported in the results. Only the kappa value was reported and no indication on reliability was stated. Please include a brief chapter on validation of the questionnaire.

Row 153: please cite the Table 2 here where results were reported. In this sentence should be stated that no significant difference were, however, noticed among the three age groups.

Row 220 to 234 – Chapter 3.6: It would be of help to include also the standardized mean differences between the compared groups. Since the SMD is independent from the sample size permit to quantify the magnitude of differences between the groups.

Row 228: the gender have a p-value of 0.54. The sentence that female genders are associated with more SEs following vaccine should be removed.

Discussion: rows 272-275: here were reported results not previously reported. Please move these to the results chapter and maintain here the discussion.

Author Response

Dear Reviewer,

We are delighted to have the opportunity to revise and resubmit our manuscript titled “Safety of ChAdOx1 nCoV-19 Vaccine: Independent Evidence from Two EU States” (Manuscript ID vaccines-1237179).

We have considered all remarks provided by all of the reviewers. Please find appended a revised version of the manuscript (with track changes highlighted) and a point-by-point rebuttal to all comments raised as detailed below. We hope our responses are satisfactory in addressing the criticisms and suggestions.

We hope the revised manuscript will be in an acceptable format. Thank you for your kind contribution.

  1. Regarding the validation of the questionnaire few details were reported in the results. Only the kappa value was reported and no indication of reliability was stated. Please include a brief chapter on the validation of the questionnaire.

Answer:

We thank the reviewer for raising this point. We have added a reference to our previous paper where we described in detail the process of development and validation and reliability testing of our instrument. Line 109-111 and Ref. 30
https://doi.org/10.3390/jcm10071428

2. Row 153: please cite Table 2 here where results were reported. In this sentence should be stated that no significant difference was, however, noticed among the three age groups.

Answer:

We sincerely thank the reviewer for this suggestion and we did implement in the results section. Line 158 and Line 160

3. Row 220 to 234 – Chapter 3.6: It would be of help to include also the standardized mean differences between the compared groups. Since the SMD is independent from the sample size permit to quantify the magnitude of differences between the groups.

Answer:

We agree with this point and we implemented the suggested changes. Line 242 - 252, and Table 6

4. Row 228: the gender have a p-value of 0.54. The sentence that female genders are associated with more SEs following vaccine should be removed.

Answer:

We agree with this point and we implemented the suggested changes. Line 229 and 231

5. Discussion: rows 272-275: here were reported results not previously reported. Please move these to the results chapter and maintain here the discussion.

Answer:

We agree with this point and we implemented the suggested changes. Line 202.

Sincerely,

Round 2

Reviewer 1 Report

Thank you for your responses.